# Hydrogen Sulfide Relaxes Human Uterine Artery via Activating Smooth Muscle BK_Ca_ Channels

**DOI:** 10.3390/antiox9111127

**Published:** 2020-11-13

**Authors:** Yan Li, Jin Bai, Yi-hua Yang, Naoto Hoshi, Dong-bao Chen

**Affiliations:** 1Department of Obstetrics & Gynecology, University of California, Irvine, CA 92697, USA; liy60@hs.uci.edu (Y.L.); baij3@hs.uci.edu (J.B.); yangyihua@gxmu.edu.cn (Y.-h.Y.); 2Department of Pharmacology, University of California, Irvine, CA 92697, USA; nhoshi@uci.edu

**Keywords:** hydrogen sulfide, BK_Ca_ channels, smooth muscle, uterine artery, women

## Abstract

Opening of large conductance calcium-activated and voltage-dependent potassium (BK_Ca_) channels hyperpolarizes plasma membranes of smooth muscle (SM) to cause vasodilation, underling a key mechanism for mediating uterine artery (UA) dilation in pregnancy. Hydrogen sulfide (H_2_S) has been recently identified as a new UA vasodilator, yet the mechanism underlying H_2_S-induced UA dilation is unknown. Here, we tested whether H_2_S activated BK_Ca_ channels in human UA smooth muscle cells (hUASMC) to mediate UA relaxation. Multiple BK_Ca_ subunits were found in human UA in vitro and hUASMC in vitro, and high β1 and γ1 proteins were localized in SM cells in human UA. Baseline outward currents, recorded by whole-cell and single-channel patch clamps, were significantly inhibited by specific BK_Ca_ blockers iberiotoxin (IBTX) or tetraethylammonium, showing specific BK_Ca_ activity in hUASMC. H_2_S dose (NaHS, 1–1000 µM)-dependently potentiated BK_Ca_ currents and open probability. Co-incubation with a Ca^2+^ blocker nifedipine (5 µM) or a chelator (ethylene glycol-bis (β-aminoethyl ether)-N,N,N′,N′-tetraacetic acid (EGTA), 5 mM) did not alter H_2_S-potentiated BK_Ca_ currents and open probability. NaHS also dose-dependently relaxed phenylephrine pre-constricted freshly prepared human UA rings, which was inhibited by IBTX. Thus, H_2_S stimulated human UA relaxation at least partially via activating SM BK_Ca_ channels independent of extracellular Ca^2+^.

## 1. Introduction

Normal pregnancy is associated with dramatically increased uterine perfusion, reflected by as high as 20–80-fold rises in uterine blood flow in the third trimester in a singleton pregnant woman [1]. Pregnancy-associated uterine vasodilation is rate-limiting for pregnancy health since rise in uterine blood flow delivers nutrients and O_2_ from the mother to fetus and exhausting CO_2_ and metabolic wastes from the fetus to mother, mandatory to support fetal development and survival. Constrained uterine blood flow has been implicated in preeclampsia, intrauterine growth restriction, and other pregnancy diseases [2,3], not only raising the morbidity and mortality of the fetus and the mother during pregnancy, but also predisposing them more susceptible to cardiovascular and other metabolic disorders later in life [4,5].

The mechanisms underlying pregnancy-associated uterine vasodilation are complex and incompletely understood; however, compelling evidence has pinpointed down a key role of locally produced vasodilators in relaxing the uterine artery (UA) smooth muscle (SM). Many vasodilators have been identified to play a role in mediating uterine vasodilation, with prostacyclin and nitric oxide as the most studied forms [6,7,8]. However, systemic inhibition of prostaglandin synthesis by indomethacin does not result in concurrent systemic or uteroplacental vasoconstriction, suggesting that uterine blood flow is not directly dependent on maintained prostaglandin synthesis [9]. Local UA NO inhibition also only modestly (≈26%) inhibits baseline pregnancy-associated uterine vasodilation [10]. These studies clearly suggest that additional mechanisms are involved to mediate pregnancy-associated uterine vasodilation.

More recently, we have reported that pregnancy augments UA production of hydrogen sulfide (H_2_S) in ewes [11] and women [12]. H_2_S has being widely accepted as the third gaseous signaling molecule of the “gasotransmitter” family that also includes nitric oxide and carbon monoxide, which exert similar pluripotent biological functions throughout the body [13]. Endogenous H_2_S is mainly synthesized by metabolizing l-cysteine via two specific enzymes, cystathionine-β-synthase (CBS) and cystathionine-γ lyase (CSE) [14]. Systemic vasculature produces H_2_S mainly via upregulating endothelial cell (EC) CSE expression or activity, which is a potent physiological vasorelaxant [15] and proangiogenic factor [16] as well as an antioxidant [13]. However, UA H_2_S production is associated with SM and EC CBS upregulation, without altering CSE expression, during pregnancy in vivo [11,12] and in cultured human UA EC in vitro [17], demonstrating that CBS is the key enzyme responsible for UA H_2_S production during pregnancy. We have also shown that a slow releasing H_2_S donor GYY4137 dose-dependently induces pregnancy-dependent UA relaxation in rats in vitro [12], suggesting that H_2_S functions as a “new” uterine vasodilator. However, how H_2_S dilates UA is currently unknown.

Activation of the ATP-sensitive potassium (K_ATP_) channels was the first mechanism demonstrated to mediate H_2_S-induced rat mesentery and aortic vasodilation [18,19]. However, local infusion of the K_ATP_ channel blocker glibenclamide does not significantly affect baseline pregnancy-associated uterine vasodilation [20]. Activation of endothelial large conductance Ca^2+^-activated voltage-dependent potassium (BK_Ca_) channels also plays a role in mediating H_2_S-induced vasodilation in rat mesenteric arteries [21]. BK_Ca_ channels are tetramer formed by the pore-forming α subunit along with regulatory β1-4 and γ1-4 subunits, which can lead to the enormous diversity in channel function [22,23]. The channel complex is activated by membrane depolarization and/or increased intracellular Ca^2+^. Opening of the channel allows K^+^ efflux leading to hyperpolarization, whereas closure of the channel causes depolarization. The activity of BK_Ca_ is critical in determining the membrane potential of vascular SM cells and hence vascular tone [24]. β1 containing BK_Ca_ channels are better characterized in SM cells, while γ subunits is newly discovered to functionally and potently regulate BK_Ca_ channel in vitro [25,26]. Pregnancy augments the expression of β1 subunit; local infusion of the BK_Ca_ blocker tetraethylammonium (TEA) abolishes pregnancy-induced UA dilation in vitro [27] and inhibits uterine blood flow in vivo [28,29,30]. Pregnancy increases UA γ1 subunit expression sevenfold and γ1 subunit deficiency results in attenuation of pregnancy-augmented increase in BK_Ca_ activity and UA dilation in mice [31]. Thus, BK_Ca_ channels play a key role in uterine hemodynamics during pregnancy.

We hypothesized herein that activation of smooth muscle cell BK_Ca_ channels mediates H_2_S-induced human UA dilation. The purpose of this study was to determine which BK_Ca_ channel subunits are expressed in human UA and cultured primary human UA SM cells (hUASMC) in vivo, as well as using primary hUASMC in culture to test (1) whether functional BK_Ca_ channels are present, and (2) whether H_2_S modifies BK_Ca_ channel activity, and if yes, by what mechanism(s). In addition, we used organ bath studies to determine if BK_Ca_ channels mediate H_2_S-induced relaxation of pressurized human UA (hUA) in vitro.

## 2. Materials and Methods

### 2.1. Ethics and Human Uterine Artery Collection

The main uterine arteries were obtained from pregnant women in the event of hysterectomy at the University of California Irvine Medical Center. Written consent was obtained from all participants, and ethical approval (IRB#2013-9763) was granted by the Institutional Review Board for Human Research at the University of California, Irvine. The tissues were collected from 5 pregnant women in an event of caesarean hysterectomy due to placenta accreta. The subjects were 26–44 years of age and at 33–37 weeks of gestation, without any other complications. The main uterine arteries were collected within 1 h after hysterectomy and placed in chilled culture medium and transported to the laboratory. Portions of each UA was allocated to be fixed in 4% paraformaldehyde or snap-frozen in liquid N_2_, and the rest was used for organ bath studies.

### 2.2. Antibodies and Chemicals

Anti-human β-actin monoclonal antibody (AM4302), anti-human BK_Ca_ γ1 subunit (PA5-38058), Dulbecco’s modified Eagle’s medium (DMEM, 12800-017), Alexa^488^ donkey anti-mouse immunoglubin G (IgG), Alexa^568^ goat anti-mouse IgG, and mounting medium containing 4′,6-diamidino-2-phenylindole (DAPI, 2105716) were from Invitrogen (San Diego, CA, USA). Anti-human BK_Ca_ β1 monoclonal antibody (sc-377023) was from Santa Cruz (Dallas, TX, USA). Anti-human BK_Ca_ γ3 monoclonal antibody (ab121412) was from Abcam (Cambridge, MA, USA). Anti-human CD31 (M0823) was from Dako (Santa Clara, CA, USA). Phenylephrine was from Tocris (Bristol, United Kingdom). Sodium hydrosulfide (NaHS, 161527), iberiotoxin (IBTX, I5904), TEA, T2265, nifedipine (N7634), ethylene glycol-bis (β-aminoethyl ether)-N,N,N′,N′-tetraacetic acid (EGTA), dithiothreitol (DTT), bovine serum albumin (BSA, A7906), fetal bovine serum (FBS, F6178), and all other chemicals were from Sigma (St. Louis, MO, USA), unless indicated.

### 2.3. Isolation and Culture of Primary UA Smooth Muscle Cells (hUASMC)

Fresh UA was washed at least 3 times with cold sterilized PBS. Connective tissues around the vessels were carefully removed and the lumen was flushed with ice-cold DMEM. After removal of EC by filling the lumen with 0.1% collagenase (type II) in phosphate-buffered saline (PBS) for 15 min at 37 °C, we cut the EC-denuded artery into ≈1 cm long rings and then soaked them in 0.05% collagenase for 20 min. The smooth muscle was then mechanically separated under a 50× stereo microscopy. The isolated smooth muscle was minced and then digested with collagenase for 30–45 min at 37 °C. Fetal bovine serum (FBS, final concentration = 10%) was added to terminate digestion. Single SM cells were collected and plated in 10 cm dishes and cultured in DMEM containing 10% FBS and 1% penicillin/streptomycin. After 7-day culture, hUASMC colonies were marked. Each colony was then picked up by using a cloning disc presoaked with 1% trypsin/EDTA as previously described [17]. Each colony was transferred into a well of a 12-well plate and cultured until ≈90% density. The cells were then stored in liquid N_2_ for experimental use within 3 passages.

### 2.4. Immunofluorescence Microscopy

Sections (6 μm) of paraffin-embedded UA rings were dehydrated and treated with proteinase K for antigen retrieval for 10 min at 37 °C, followed by rinsing 3 times with PBS. After incubation with 1% bovine serum albumin (BSA) in PBS to block nonspecific binding for 30 min at room temperature, the sections were incubated with anti-human BK_Ca_ β1 (1:50) or γ1 (1:50) subunit at 4 °C overnight. IgG was used as negative control. All antibody incubations were performed in 0.5% BSA/PBS. The sections were washed 3 × 10 min with PBS, and then incubated with Alexa^568^ mouse immunoglobulin (IgG, 1:1000) for 1 h at room temperature. After 3 × 10 min washing with PBS, the sections were blocked with 1% BSA/PBS for 30 min at room temperature. The sections were incubated with anti-human CD31 (1:200) at 4 °C overnight, washed, and then incubated with Alexa^488^ anti-mouse IgG (1:1000) for 1 h at room temperature. The sections were washed and then mounted with anti-fade mounting medium containing DAPI. Sections were examined under a confocal laser scanning microscope (Olympus SV3000) and images were acquired for quantifying levels of BK_Ca_ subunits (mean red fluorescence intensity) in SMC and EC as previously described [12].

### 2.5. RNA Extraction and Reverse Transcription Polymerase Chain Reaction (RT-PCR)

Total RNAs were extracted from the main UA tissue (≈100 mg) or cultured hUASMC (≈2 × 10^5^ cells) using Trizol reagent (Invitrogen, Carlsbad, CA, USA) and quantified by OD_260/280._ Complementary DNA was synthesized by reverse transcription with random primers and AMV Reverse Transcriptase (Promega, Madison, WI, USA) and then used for detecting mRNAs of BK_Ca_ subunits by PCR with gene-specific primers as listed in Table 1. PCR was run as follows: 95 °C for 5 min, followed by 38 cycles of 95 °C for 30 s, 62 °C for 30 s, and 72 °C for 30 s, and then 72 °C for 5 min and 4 °C. The amplicons were confirmed by sequencing.

### 2.6. Western Blot

UA and cultured hUASMC proteins were extracted using a lysis buffer as previously described [32]. Equal amounts of total protein extracts (20 μg/lane) were separated on 10–15% SDS-PAGE and transferred to polyvinylidene difluoride membrane. Proteins were determined by immunoblotting with antibodies against anti-human BK_Ca_ β1 (1:100) or γ1 (1:200) subunits in Tris-buffered saline (TBS) containing 5% BSA as described previously [12]. β-actin was determined as a control for sample loading.

### 2.7. Electrophysiology

Electrophysiological experiments were performed as described previously [33,34]. Briefly, cultured primary hUASMC were used for whole-cell, inside-out, and outside-out recordings with an Axonpatch-200B connected to a Digidata 1322A using pClamp10 software (Molecular Devices, CA, USA). The patch pipettes were fabricated from borosilicate glass (Havard Apparatus) and had electrode resistances from 2–4 MΩ with an access resistance from 3–10 MΩ. Cells with current leakage less than 100 pA in the whole-cell mode were selected for analysis. Sampling frequencies for whole-cell current and single-channel recordings were 1 kHz and 5 kHz, respectively. Data were filtered with a low-pass 4-pole Bessel filter set at 1 kHz, which results in a 10–90% rise time of 350 μs. For whole-cell and outside-out single-channel recordings, the bath solution contained (mM) 144 NaCl, 5 KCl, 2 CaCl_2_, 0.5 MgCl_2_, 10 2-[4-(2-hydroxyethyl)piperazin-1-yl]ethanesulfonic acid (HEPES), and 10 glucose, at pH 7.4 adjusted with 10N NaOH. The recording pipette solution contained (mM) 140 KCl, 1 MgCl_2_, 5 Na_2_ATP, 5 EGTA, and 2.5 CaCl_2_, at pH 7.2. The final free Ca^2+^ concentration was calculated by the Webmaxc extended calculator (http://www.stanford.edu/~cpatton/webmaxcE.htm) and estimated to be 10 μM in the control pipette solution, which was adjusted for indicated free Ca^2+^ concentration in the text by changing CaCl_2_ concentration or by adding EGTA.

BK_Ca_ channels keep open with intracellular free Ca^2+^ higher than 50 µM [35], making it hard to qualify the channel activity with a continuous high free Ca^2+^ level. Thus, we performed all the tests with intracellular free Ca^2+^ no higher than 10 µM. Ca^2+^-free recordings were performed with the same bath solution containing 5 mM EGTA. Channel blockers were added into the bath solutions unless stated otherwise. For inside-outside single-channel recordings, the pipette and the bathe solutions are the same as the pipette solutions of whole-cell recordings as described above. Test solutions were applied via a gravity-driven system controlled by VCS-66MCS (Warner Instrument, Hemden, CT, USA). For rapid solution exchange (≈300–500 ms), we held membrane patches in a stream of the experimental solution from a second pipette. Single-channel current amplitudes were calculated by fitting amplitude histograms to a Gaussian distribution. Channel open probability was expressed as *P*_open_ = NPo/n, where NPo = [(to)/(to + tc)]. *P*_open_ = open probability for one channel; to = sum of open times; tc = sum of closed times; N = actual number of channels in the patch; and n = maximum number of individual channels observed in the patch. Experiments were repeated at least 3 times and data were calculated as the mean ± SEM (standard error of the mean). The linear regression is shown in the single channel current-voltage (I-V) curve. *P*_open_ was fit with Gaussian function. Single-channel conductance (g, pico Siemens, pS) was calculated using I/U; I = single-channel current (pA), U = membrane potential (mV).

The whole-cell patch-clamp technique was used to record K_ATP_ channel currents as previously described [18]. The bath solution for recording whole-cell K_ATP_ current contained (mM) 140 NaCl, 5.4 KCl, 1.2 MgCl_2_, 10 HEPES, 1 EGTA, and 10 glucose, with pH adjusted to 7.4 with NaOH. The pipette solution contained (mM) 140 KCl, 1 MgCl_2_, 10 EGTA, 10 HEPES, 5 glucose, 0.3 Na_2_ATP, and 0.5 MgGDP, with pH adjusted to 7.2 with KOH. Cells were superfused continuously with the bath solution at a rate of approximately 2 mL/min. Solution change in the recording chamber was accomplished within 30 s.

All patch clamp recordings were carried out at room temperature (20–22 °C). NaHS was used as a source of H_2_S; working solutions were prepared immediately before use as H_2_S gas evaporates 10–15% from the solution within 30 min at 37 °C [36]. Stock solution of nifedipine was dissolved in DMSO; the final DMSO concentration did not exceed 0.05%, which did not change the currents in control experiments.

### 2.8. Organ Bath Studies

Freshly prepared UA rings (2–5 mm in length) were placed in ice-cold Krebs–Ringer bicarbonate (KRB) bath solution containing (mM) 118.5 NaCl, 4.75 KCl, 1.2 MgSO_4_, 1.2 KH_2_PO_4_, 25 NaHCO_3_, 2.5 CaCl_2_, and 5.5 glucose, with pH 7.4 adjusted with HCl. The UA rings were mounted onto a tension transducer (JZJ01H) under a stable resting tension in organ bath chambers containing 5 mL of KRB solution at 37 °C, gassed with 95% O_2_ and 5% CO_2_. The rings were allowed to equilibrate for at least 30 min, with chamber solution changed every 15 min. Endothelium integrity was determined by response to 10 μM acetylcholine as previously described [12]. Only endothelium-intact rings were used, which were preload with a tension at 1.5 g after equilibration; contraction was recorded when the tension was stable for at least 15 min. Rings were pre-contracted with 10 µM phenylephrine. Rings rapidly responding to phenylephrine in 5 min with more than 2 mN contraction were selected for recording the dose–response relaxation curves of NaHS in the presence or absence of the selective BK_Ca_ channel blockers. Each drug was allowed at least 5 min to respond. Changes in the isometric tension were recorded and analyzed with a Multiple Channel Physiology Signal Recording System (RM-6240EC, Chengdu Instrument Factory, Chengdu, China).

### 2.9. Statistics

Results are expressed as means ± standard error. Significant levels were determined by using the paired Student’s *t*-test or one-way ANOVA followed by Bonferroni test for multiple comparisons, whichever appropriate, using GraphPad Prism 8. Significant difference was accepted at *p* < 0.05.

## 3. Results

### 3.1. Expression of BK_Ca_ Channels in UA In Vitro and Primary UASMC In Vitro

BK_Ca_ channels are tetramer formed by the pore-forming α subunits, along with the regulatory β1–4 and γ1–4 subunits [22,23]. By using RT-PCR and sequencing conformation, we detected α, β1, β3, β4, and γ1–3, but not β2 and γ4, mRNAs in pregnant human UA and cultured primary hUASMC (Figure 1A). Since β1, γ1, and γ3 subunits are the most important ones for mediating UA adaptation to pregnancy [27,31,37], we further examined their proteins in human uterine arteries and cultured hUASMC. We tested two commercially available antibodies against γ1 and γ3 subunits to detect their protein levels by Western blot and immunofluorescence microscopy. The γ1 subunit was only detectable by Western blot with one antibody (PA5-38058) but not by immunofluorescence microscopy, whereas γ3 subunit was detectable by immunofluorescence microscopy with the Abcam antibody (ab121412) but not by Western blot with all other commercial antibodies. Immunoblotting detected β1 and γ1 proteins in both UA and cultured hUASMC and they did not change in three passages (Figure 1B). Immunofluorescence microscopy analysis revealed that both VSM and EC expressed β1 and γ3 proteins; however, levels of both β1 and γ3 proteins in SM cells were significantly greater than that in the CD31^+^ EC. In addition, histological analysis showed that both β1 and γ3 proteins are not expressed in all ECs as β1 or γ3 proteins were only found in some regions of the CD31^+^ EC linings (Figure 1C).

### 3.2. Functional BK_Ca_ Channels in Primary hUASMC In Vitro

To determine if BK_Ca_ channels were functional in cultured hUASMC, we introduced whole-cell and single-channel patch clamp with the selective BK_Ca_ channel blockers: iberiotoxin (IBTX, 100 nM) or low concentration of TEA (1 mM). Ion currents were elicited in response to a series of voltage pulses from −60 mV holding potential to +80 mV in steps of 10 mV. Both IBTX and TEA blocked the outward current significantly compared with the baseline holding membrane potential from +40 mV to +80 mV (*p* < 0.05, Figure 2A–C). In the inside-out patch, cultured hUASMC BK_Ca_ channels showed a single-channel conductance of 201 ± 19.08 pS (*n* = 8) in a symmetrical high K^+^ solution (140 mM) on both sides of the cell membrane, which was consistent with reported values [38] (Figure 2D,E). In outside-out/inside-out single-channel recording with 100 nM free Ca^2+^ in the pipette solution at +40 mV holding membrane potential, the observed single-channel activities were blocked by IBTX or TEA, confirming the observed 200 pS channels to be BK_Ca_ channels (Figure 2F). Open probability (*P*_open_) of the channels was decreased from 0.04 ± 0.009 (*n* = 10) to 0.0019 ± 0.00046 (*n* = 5, *p* < 0.05) by IBTX, and to 0.0026 ± 0.0011 (*n* = 5, *p* < 0.05) by TEA. These results indicate the presence of IBTX- and TEA-sensitive functional BK_Ca_ channels in hUASMC in vitro.

To determine if BK_Ca_ channels were functional in cultured hUASMC, we introduced whole-cell and single-channel patch clamps with the selective BK_Ca_ channel blockers, IBTX (100 nM) and low concentration of TEA (1 mM), separately. Ion currents were elicited in response to a series of voltage pulses from −60 mV holding potential to +80 mV in steps of 10 mV. Both IBTX and TEA significantly blocked the outward current in comparison with the baseline holding membrane potential from +40 mV to +80 mV (*p* < 0.05, Figure 2A–C). In the inside-out patch, cultured hUASMC BK_Ca_ channels showed a single-channel conductance of 201 ± 19.08 pS (*n* = 8) in a symmetrical high K^+^ solution (140 mM) on both sides of the cell membrane (Figure 2D,E). With 100 nM free Ca^2+^ in the pipette solution at +40 mV holding membrane potential, the single-channel BK_Ca_ currents were blocked by IBTX or TEA (Figure 2F). *P*_open_ of BK_Ca_ decreased significantly from 0.04 ± 0.009 (*n* = 10) to 0.0019 ± 0.00046 (*n* = 5, *p* < 0.05) by IBTX, and to 0.0026 ± 0.0011 (*n* = 5, *p* < 0.05) by TEA, indicating the presence of IBTX- and TEA-sensitive functional BK_Ca_ channels in primary hUASMC in vitro.

### 3.3. H_2_S Increased Ca^2+^-Activated and Voltage-Dependent K^+^ Currents in hUASMC

When sodium hydrosulfide (NaHS) was applied to the extracellular solution, it rapidly dissociated into Na^+^ and HS^−^, and HS^−^ associated with H^+^ to produce H_2_S. However, only the H_2_S molecule, but not HS^−^, is able to pass the plasma membrane, as H_2_S possess approximately fivefold greater lipophilic solubility than water [39]. Addition of NaHS (100 µM) caused a significant and reversible increase of membrane outward currents, and current voltage relationships were obtained within 1–3 min after NaHS incubation. NaHS on BK_Ca_ activity was assessed with whole-cell and single-channel recordings. NaHS significantly augmented the whole-cell outward current from 60 mv membrane potential (*p* < 0.05, Figure 3A–C), which was sensitive to 1 mM TEA (*p* < 0.05, Figure 3A–C), indicating that the augmented outward currents were BK_Ca_-mediated. In single-channel recordings, NaHS increased *P*_open_ from baseline (0.1258 ± 0.01) to 0.3107 ± 0.02, and standard bath solution reversed the NaHS-induced *P*_open_ to 0.1533 ± 0.01; most of the outward currents were sensitive to 1 mM TEA (*p* < 0.05, Figure 3A–C). With 10 µM free Ca^2+^ in the pipette solution at +40 mV holding membrane potential, NaHS increased *P*_open_ of BK_Ca_ from 0.468 ± 0.04226 to 0.7742 ± 0.02664 (*p* < 0.01). The H_2_S-induced *P*_open_ of BK_Ca_ was also observed at lower holding potentials from −10 mV to + 20 mV (*n* = 6, *p* < 0.05 vs. baseline, Figure 3F). NaHS stimulated BK_Ca_ activity in a U-shaped concentration-dependent manner; NaSH at 100 and 500 μM significantly increased *P*_open_ of BK_Ca_ channels by 166.6 ± 29% and 198.1 ± 35% (*n* = 10), respectively. Low (10 µM) and high (1 mM) concentrations of NaHS also increased *P*_open_ by 134.9 ± 24% and 160.2 ± 62% (*n* = 10), but these responses did not differ statistically from the controls (Figure 3G).

### 3.4. H_2_S Activation of hUASMC BKCa Was Independent of Extracellular Ca^2+^

Voltage and cytosolic Ca^2+^ are the two major regulatory components physiologically for BK_Ca_ channels [40,41]. To analyze whether H_2_S-induced activity in hUASMC BK_Ca_ depends on voltage and cytosolic Ca^2+^, we determined the effects of extracellular and intracellular Ca^2+^ on the NaHS (100 µM)-induced *P*_open_ of BK_Ca_, holding at different membrane potentials from −60 mV to +80 mV. The representative traces showed *P*_open_ in response to voltage ramp in control and NaHS groups (Figure 4A). Following NaHS treatment, *P*_open_ of BK_Ca_ increased by 137.8 ± 24% at 10 mV, 181.4 ± 17% (*p* < 0.05) at +20 mV, and 237 ± 57% (*p* < 0.05) at +40 mV. The increases in the NaHS-induced *P*_open_ of BK_Ca_ were less effective when holding potentials were higher than +50 mV, by 161.5 ± 22%, 146.7 ± 24%, 139.4 ± 14%, at +60 mV, +70 mV, and +80 mV, respectively (Figure 4B). When the holding pipette solution Ca^2+^ concentrations were 0, 0.1, and 10 μM, the NaHS-induced *P*_open_ of BK_Ca_ increased by 193 ± 39% (*p* < 0.05, *n* = 4), 172 ± 26% (*p* < 0.05, *n* = 5), and 150.5 ± 14% (*n* = 6, *p* < 0.05), respectively (Figure 4C). NaHS also induced comparable significantly increased *P*_open_ of BK_Ca_ from holding potential of +10 mV to +80 mV when thylene glycol-bis(β-aminoethyl ether)-N,N,N′,N′-tetraacetic acid (EGTA, 5 mM) was added in the bath solution, which should eliminate free Ca^2+^ (Figure 4D). When a Ca^2+^ channel blocker nifedipine (5 µM) was applied, it also did not affect the NaHS-induced *P*_open_ of BK_Ca_ (Figure 4E).

### 3.5. H_2_S-Induced BK_Ca_ Activation Is Redox-Sensitive

The activity of BK_Ca_ channels depends on the redox state of the sulfhydryl groups in the channel proteins [42,43,44], and oxidation reduces BK_Ca_ activity [45,46]. To study if the NaHS-induced BK_Ca_ activation is redox-dependent, we determined the effects of a reducing agent dithiothreitol (DTT, 1 mM) added into the bath solution on the NaHS-induced *P*_open_ of BK_Ca_. Treatment with NaHS increased *P*_open_ of BK_Ca_ from baseline 0.036 ± 0.011 to 0.119 ± 0.032 (*p* < 0.05); co-incubation with DTT decreased the NaSH-induced *P*_open_ of BK_Ca_ to 0.072 ± 0.034 (*p* < 0.05) (Figure 5A,B). Co-incubation with DDT blocked NaHS-induced BK_Ca_ activation; however, this effect was rapidly diminished and then all channel activities were blocked (Figure 5C). DTT alone did not alter BK_Ca_ channel P_open_ (Figure 5B,C).

### 3.6. H_2_S Relaxed Human UA via BK_Ca_ Channel

Incubation with increasing concentrations (1, 10, 100, 500 µM) NaHS stimulated dose-dependent relaxation of freshly prepared human UA rings that were pre-constricted with 10 µM phenylephrine (Figure 6A). Pretreatment with the selective BK_Ca_ channel inhibitor IBTX (100 nM) blocked the NaHS-induced UA relaxation (Figure 6B).

### 3.7. H_2_S Did Not Activate K_ATP_ Channels in hUASMC

Since K_ATP_ channels are direct effectors of H_2_S [36,47,48], we determined whether H_2_S activates K_ATP_ channels in hUASMC. Treatment with NaHS (300 μM) [18] did not alter baseline inward currents stimulated by 140 mM K^+^, indicative of K_ATP_ channel activity (Figure 7A); however, co-incubation with the K_ATP_ channel blocker glibenclamide (10 μM) inhibited K_ATP_ channel activity (Figure 7A,B).

## 4. Discussion

Consistent with the well-documented vasodilatory effect of H_2_S in many systemic arteries [15,36,49,50], we were the first to report that H_2_S dilates pressurized UA in a pregnancy- and vascular bed-dependent manner in rats [12]. The current study demonstrates for the first time that H_2_S activates BK_Ca_ channels in hUASMC, as well as the fact that incubation of the specific BK_Ca_ channel blocker IBTX completely blocks H_2_S-induced relaxation of pre-constricted human UA rings in vitro. These findings provide direct evidence for a role of smooth muscle BK_Ca_ channels in mediating the vasodilatory effects of H_2_S in the UA, further supporting the notion that H_2_S is a novel UA vasodilator.

Endogenous H_2_S is a gaseous signaling molecule that is mainly synthesized by CBS and CSE in various human tissues, while other enzymes such as 3-mercaptopyruvate sulfurtransferase (3MST) in combination with cysteine aminotransferase (CAT) may also play a role [51]. Our recent studies have consistently shown that H_2_S production is upregulated in the UA via selectively upregulating EC and SM CBS expression, without altering the expressions of CSE, 3MST, and CAT in vivo [11,12,32] and in human UAEC in vitro [17]. In this study, NaHS was used as a source of H_2_S. In aqueous solution, NaHS dissociates to Na^+^ and HS^−^, and HS^−^ associates with H^+^ to produce H_2_S. In neutral solution, one-third of NaHS exists as H_2_S, and the remaining two-thirds are present as HS^−^ [52]. Thus, the solution of H_2_S is about ≈66% of the original concentration of NaHS [53]. The liberation of <1 mM Na^+^ from NaHS is negligible since the bath solution contained 145 mM Na^+^. The concentrations of NaSH used in this study ranged from 1 to 1000 µM, which did not change the pH of the buffered solution. The concentration of NaSH used in most of the experiments was 100 µM, equivalent to ≈60 µM H_2_S, which is close to the physiological plasma levels (less than ≈50 µM) of H_2_S in humans [51]. Our data show that addition of 100 µM NaSH significantly activated BK_Ca_ channels in hUASMC and dilated human UA rings in vitro, showing that H_2_S is a physiological UA dilator.

Activation of K_ATP_ channels was the first mechanism that has been shown to mediate H_2_S-induced vasodilation in rat mesentery artery [19], which has been confirmed by many follow-up studies in other vessels [36,47,48]. However, activation of K_ATP_ accounts for no more than half of the effect of H_2_S to relax most vessels [54]. Likewise, opening of BK_Ca_ channels results in K^+^ efflux, causing membrane hyperpolarization of vascular SMC as a key mechanism for vasodilation [40]. UA BK_Ca_ activity increases in pregnant sheep [55]. Local infusion of TEA to block BK_Ca_ channels abolishes pregnancy-induced UA dilation in vitro [27] and inhibits pregnancy-associated uterine blood flow in vivo [28,29,30], while local infusion of glibenclamide to block the K_ATP_ channels does not significantly affect baseline pregnancy-associated uterine blood flow [20]. Consistently, we did not observe a significant effect of H_2_S on K_ATP_ channels in hUASMC. Why H_2_S, unlike other systemic SMCs, does not activate K_ATP_ channels in hUASMC warrants further elucidation. Nonetheless, our current data, along with data from in vivo studies using blockers of various K^+^ channels to determining their role in pregnancy-associated rise in uterine blood flow [20,28,30], suggest that activation of SM BK_Ca_ channels is important for mediating H_2_S-induced UA dilation.

BK_Ca_ channels, also known as BK/MaxiK/Slo1/K_Ca_1.1 channels, are K^+^ channels of largest single-channel conductance (≈200–300 pS) [55]. The essential structure of BK_Ca_ channels consist of the α-ubunit and can be complemented with the regulatory subunits, including the β isoforms (1–4) and γ isoforms (1–4) [56,57]. The β1 subunit is essential for increasing voltage sensitivity when intracellular free Ca^2+^ is beyond 1 µM [22,58]. The γ1–γ4 are auxiliary subunits that greatly modify channel activity in mammalian cells [25,26,59,60,61]. The expression and their physiological and pathological functions of SM BK_Ca_ channels have been well studied in other tissues in mammalians [62], but their distribution and function remains to be understudied in UA smooth muscle cells (UASMC). Previous studies have shown SM expression of α and β1 [63] and γ1 [31] subunits in UA; the α subunit is constitutively expressed and the β1 and γ1 subunits are significantly upregulated in pregnancy [31,55]. Herein, we show the expressions of α, β1, β3, β4, and γ1–3, but not β2 and γ4, mRNAs, and β1 and γ1 and γ3 proteins in hUA and cultured hUASMC. Which subunit(s) of these isoforms are responsible for the H_2_S-induced BK_Ca_ activity in hUASMC? Our current study did not provide any data to address this important question; however, β1-containing BK_Ca_ channels are sensitive to IBTX and low concentration of TEA [22,64]; the similar pharmacological properties with IBTX and TEA obtained in this study has implicated a functional role of β1 subunit in H_2_S-induced BK_Ca_ activity in hUASMC, consistent with previous studies showing that β1 subunits are upregulated and are important for increasing SM BK_Ca_ activity in the UA in response to estrogen stimulation and during pregnancy [55,65]. The γ1 subunit containing BK_Ca_ channels are featured by the ≈120 mV leftward shift at 0 and elevated cytosolic Ca^2+^, which facilitates BK_Ca_ channel activity [22]; the γ3 is less studied but also related to Ca^2+^ sensitivity of the channel [25]. γ1 subunit is upregulated sevenfold in mouse UA in pregnancy [31]. Future studies are warranted to delineate whether they are involved in the H_2_S-induced UASMC BK_Ca_ activity since γ1 and γ3 proteins are highly expressed in hUA and retained in hUASMC in culture.

How does H_2_S activate BK_Ca_ channels in hUASMC? With BK_Ca_ channels being Ca^2+^-activated and voltage-dependent ion channels, activation requires either elevation of intracellular Ca^2+^ or depolarization of cell membrane [66]. The free intracellular Ca^2+^ concentration under resting conditions is ≈150 nM, although it is oscillating in some cells, and can increase as high as 500 nM [67]. In addition, Ca^2+^ concentrations in the vicinity of BK_Ca_ channels after influx through Ca^2+^ channels are between 4 and 30 μM [68], which are dramatically higher compared to average cytoplasmic free internal Ca^2+^ concentrations. Free internal Ca^2+^ concentrations used in our experiments are within this range. In resistance-sized cerebral arteries, ryanodine receptor-sensitive Ca^2+^ sparks in sarcoplasmic reticulum (SR) activate BK_Ca_ channels [69], while in the resting state of cerebral artery activation of BK_Ca_ channels relies on Ca^2+^ influx through L-type voltage-dependent calcium channels (LTCC) [70]; however, this is not the case in coronary or mesenteric arteries, indicating that different mechanisms for BK_Ca_ channel activation varies among vessels from different vascular beds. In hUASMC, blockade of LTCC using nifedipine does not affect H_2_S-induced BK_Ca_ activity recorded by whole-cell patch clamp, suggesting LTCC-mediated Ca^2+^ influx is not involved. Similar results were also obtained with 0 free Ca^2+^ bath solution containing EGTA, indicating that H_2_S-induced BK_Ca_ activity is independent of extracellular Ca^2+^, sharing similar properties with the H_2_S-responsive BK_Ca_ channels in rat pituitary tumor cells [71]. In ovine UASMC, recent studies have shown that ryanodine-receptor sensitive Ca^2+^ sparks are important for pregnancy and estrogen stimulation of BK_Ca_ channel activity [72]. In rat mesenteric arteries, H_2_S-induced vasodilation requires activation of endothelial BK_Ca_ channels and smooth muscle Ca^2+^ sparks [21]. Thus, future studies are needed to determine if SR Ca^2+^ sparks mediate activation of the H_2_S-induced BK_Ca_ channels in UASMC.

Apart from Ca^2+^ and voltage, many other mechanisms are also involved in regulating BK_Ca_ channel activity, including phosphorylation by protein kinases such as protein kinase A (PKA), PKG, and PKC; PKA and PKG activate BK_Ca_ channels through modulating the channel kinetics, while PKC shows an inhibitory manner on the channels [66]. In the present study, we show that NaHS modulates BK_Ca_ channels directly by using outside-out single-channel patch recording. In the whole-cell patch recording mode, NaHS may modulate BK_Ca_ channel activity indirectly through protein kinase-mediated phosphorylation. However, this idea needs to be further explored. In addition, direct sulfhydrating proteins in reactive cysteines has been recently recognized to be a major mechanism for H_2_S to elicit its biological functions [73]. Direct sulfyhydration of Kir 6.1 on C43 has been shown to be a key mechanism for H_2_S-induced K_ATP_ channel activation [74]. In this study, the H_2_S-response BK_Ca_ channel was found to be sensitive to DTT, which completely prevents protein cysteine modifications including sulfhydration [73]. Thus, this mechanism is highly likely involved in H_2_S-induced BK_Ca_ channel activation in hUASMC, although detailed mechanisms around sulfhydration in terms of which subunit(s) and on which specific cysteine(s) are involved are still to be determined.

## 5. Conclusions

Altogether, we have shown herein that functional BK_Ca_ channels are present in human UASMC, which can at least partially mediate the vascular relaxation effects of H_2_S in human UA in vitro. However, it is necessary to point out that research in H_2_S in uterine hemodynamics is still at a very early stage. Future studies are warranted to address many important questions so that a physiological and pathophysiological role of H_2_S and the underlying mechanisms in uterine hemodynamic regulation can be delineated, pertaining to normal pregnancy and hypertension-related pregnancy complications such as preeclampsia.

## Figures and Tables

**Figure 1 antioxidants-09-01127-f001:**
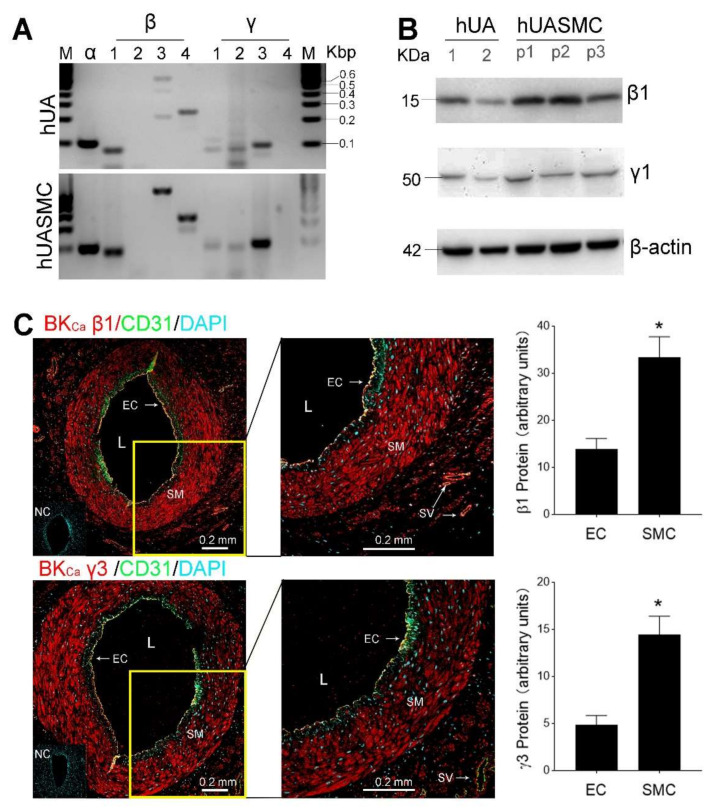
BK_Ca_ channel expression in human uterine artery. (**A**) Expression of mRNAs of BK_Ca_ subunits in human uterine artery (hUA, upper panel) and cultured primary hUA smooth muscle cells (hUASMC, lower panel). Steady-state mRNAs of α, β1–4, and γ1–4 subunits were detected by RT-PCR. The amplicons were sequencing confirmed. M, 100 bp DNA ladder. (**B**) β1 and γ1 proteins detected by immunoblotting in hUA from two women and primary hUASMC in three passages (P). (**C**) Localization of β1 and γ3 proteins by immunofluorescence microscopy. SMC; smooth muscle cells; EC; endothelial cells; L; lumen; NC: negative control; SV; small vessels. Graph summarized levels of EC and SMC β1 and γ3 proteins (*n* = 3). * *p* < 0.05.

**Figure 2 antioxidants-09-01127-f002:**
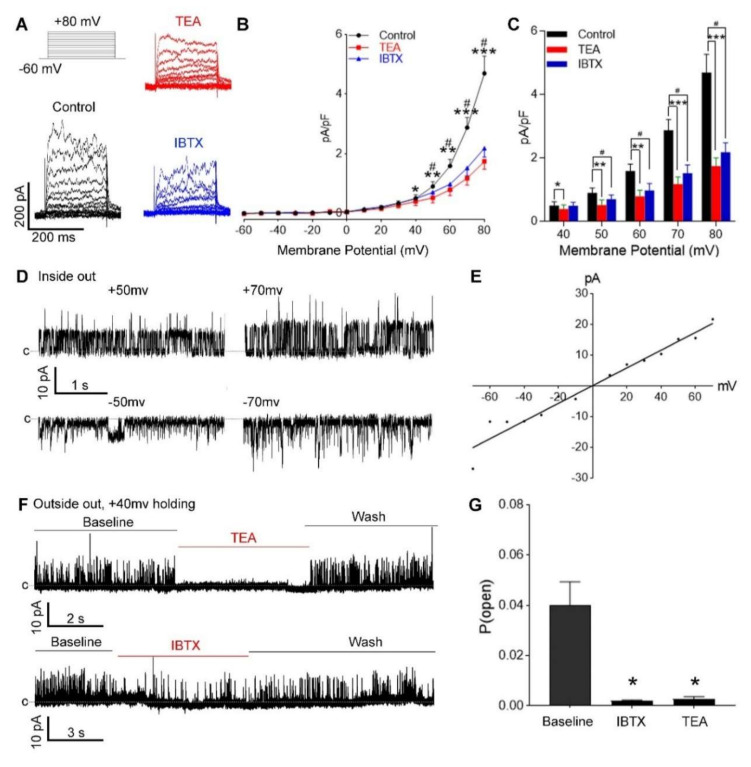
Functional BK_Ca_ channels in primary hUASMC in vitro. (**A**) The top-left figure shows voltage-triggered protocol of the whole-cell patches, in which cells were held at −60 mV followed by a 10-mV voltage increment until +80 mV. Representative voltage-dependent current sweeps of cultured hUASMC in control (black), tetraethylammonium (TEA) (1 mM, red), and iberiotoxin (IBTX) (100 nM, blue) groups. (**B**,**C**) Current density was used to quantify channel activity, illustrated as current/capacitance (pA/pF). Both TEA (red) and IBTX (blue) inhibited ion currents significantly from holding potential of +40 mV to +80 mV. # *p* < 0.05, * *p* < 0.05; ** *p* < 0.01; *** *p* < 0.001; IBTX or TEA vs. control. (**D**,**E**) Inside-out patch of cultured hUASMC with symmetrical 140 mM K^+^ showed outward currents with holding potential of +50 and +70 mV (upper panel in (**D**)), and inward currents at −50 and −70 mV (lower panel in (**D**)), in which a conductance of ≈250 pS in the representative trace indicates the presence of big conductance K^+^ channels. (**F**) The big conductance K^+^ channels were sensitive to TEA (upper panel) and IBTX (lower panel) in outside-out patch with 100 nM free Ca^2+^ in pipette solution. (**G**) Open probability (*P*_open_) was used to quantify BK_Ca_ activity. * *p* < 0.05 vs. baseline. c indicates the close state of channels.

**Figure 3 antioxidants-09-01127-f003:**
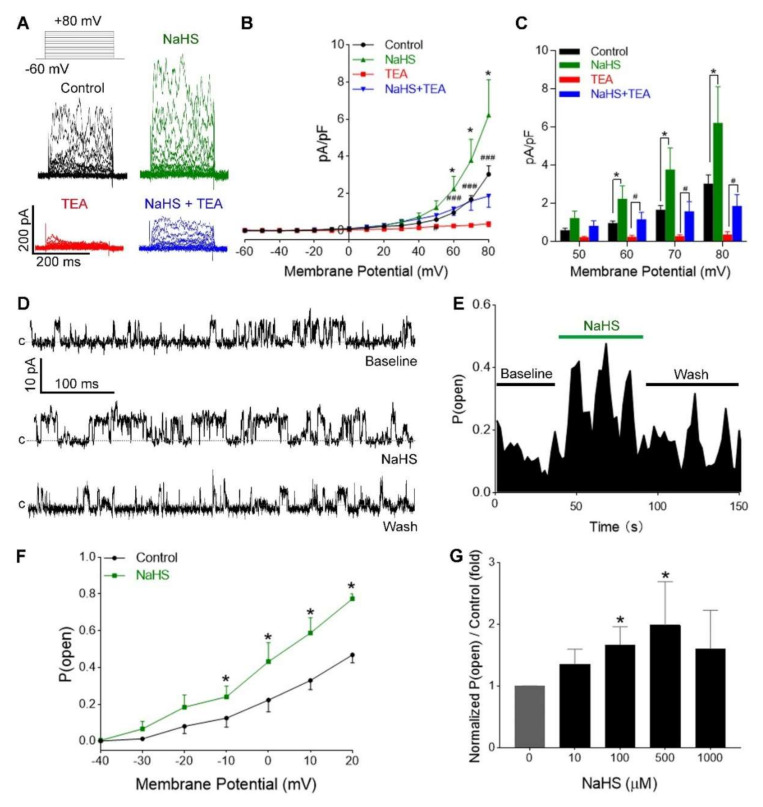
H_2_S activation of BK_Ca_ in hUASMC. (**A**) Representative sweeps of voltage-dependent currents in control (black), TEA (red), H_2_S doner NaHS (green), and NaHS + TEA (blue) groups. NaHS increased whole-cell currents (green), and TEA blocked NaHS-induced currents (blue). (**B**,**C**) Current densities in control (*n* = 10), NaHS (*n* = 10), TEA (*n* = 5), and NaHS + TEA (*n* = 5) groups. * *p* < 0.05, NaHS vs. control; ### *p* < 0.001, TEA vs. control. In addition to TEA-sensitive channels, NaHS also activated TEA-insensitive channels. # *p* < 0.05 TEA vs. NaHS + TEA. (**D**,**E**) Representative outside-out single-channel currents of BK_Ca_ in baseline, NaHS, and washout with standard bath solutions at holding potential of +40 mV and with 10 μM free Ca^2+^ in the pipette solution. (**F**) *P*_open_ of BK_Ca_ was also augmented in lower membrane potentials of −10 mV to +20 mV (*n* = 6 in control and *n* = 4 in NaHS groups). * *p* < 0.05, vs. control. (**G**) Dose–response of NaHS on BK_Ca_ channel activity. *n* = 10/group. *c:* indicates the close state of channels.

**Figure 4 antioxidants-09-01127-f004:**
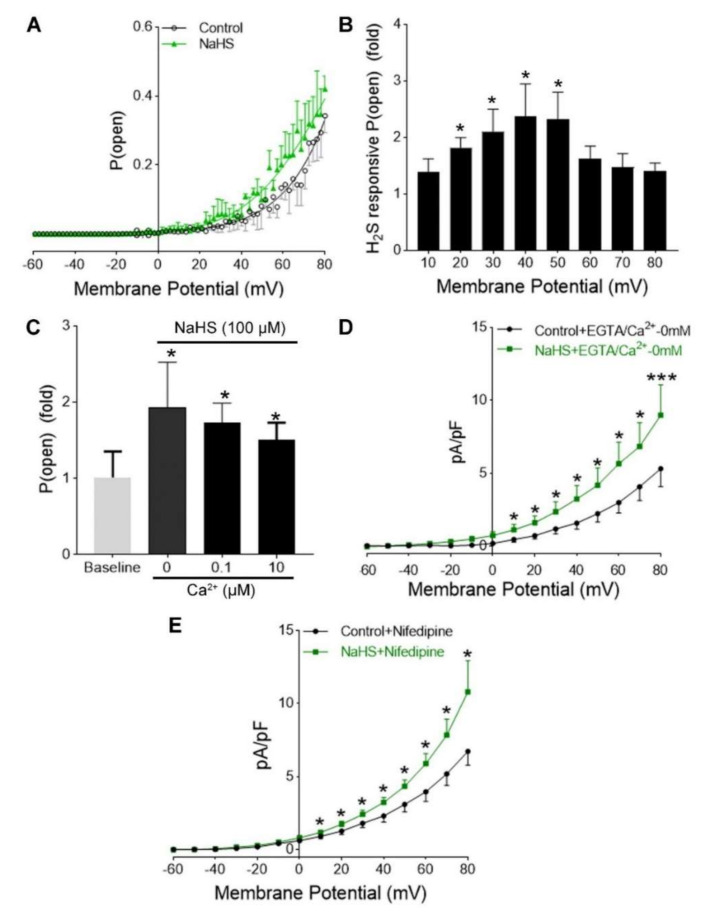
Properties of H_2_S-responsive BK_Ca_ in hUASMC in vitro. (**A**) Open probability (*P*_open_) of BK_Ca_ recorded with membrane potential of −60 mV to +80 mV in outside-out single-channel mode. (**B**) *P*_open_ with different membrane potentials with and without 100 μM NaHS. NaHS increased *P*_open_ with membrane potential of +20 to +50 mV. * *p*< 0.05, NaSH vs. control at each membrane potential (*n* = 8). (**C**) Ca^2+^ concentrations (0, 0.1, and 10 μM) in pipette solution on *P*_open_ of BK_Ca_ channels in response to NaHS at −60 mv membrane potential. * *p* < 0.05 vs. baseline. (**D**) Bath solution with 5 mM EGTA on current density in response to NaHS at −60 mv membrane potential. NaHS (*n* = 11) increased current density in free Ca^2+^ bath solution. * *p* < 0.05, *** *p* < 0.001 vs. control; *n* = 11/group. (**E**) Effects of nifedipine (5 μM) in bath solution on NaHS (100 μM)-induced current density. * *p* < 0.05 vs. control; *n* = 6/group.

**Figure 5 antioxidants-09-01127-f005:**
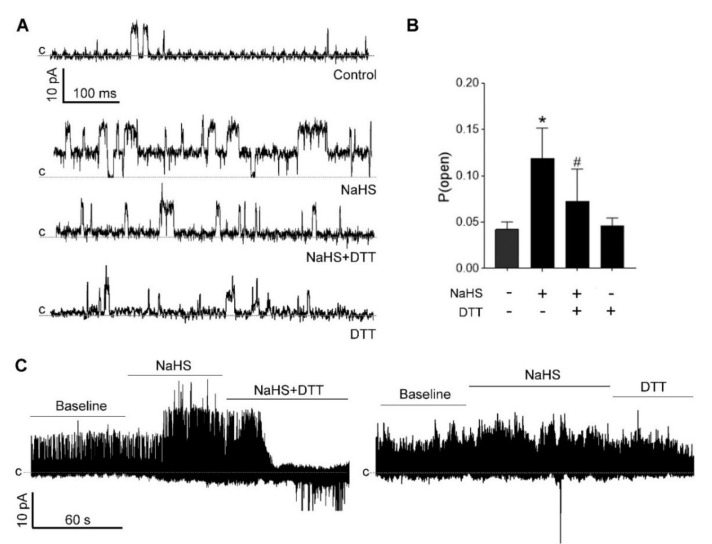
The BK_Ca_ channel opening activity of H_2_S was redox-sensitive. (**A**) Original outside-out single-channel currents with dithiothreitol (DTT, 1 mM) in the pipette solution before and during NaHS (100 μM) application. Holding potentials were +60 mV with 0.1 μM free Ca^2+^ in the pipette solution. (**B**) Open probability (*P*_open)_ of BK_Ca_ was significantly increased from 0.036 ± 0.011 (*n* = 7) in the control group to 0.119 ± 0.032 (*n* = 7) by 100 µM NaHS; the addition of DTT decreased the H_2_S-induced *P*_open_ to 0.072 ± 0.034 (*n* = 6). * *p* < 0.05 compared with control group; # *p* < 0.05 compared with NaHS group. *P*_open_ was 0.046 ± 0.008 (*n* = 4) in the presence of DTT alone. (**C**) DTT on NaHS (100 μM) stimulated outside-out single-channel currents. Co-incubation with DTT (1 mM) blocked the outward currents (left), while DTT alone did not alter baseline outward current (right). Currents represent similar experiments from different cells. *c:* indicates the close state of channels.

**Figure 6 antioxidants-09-01127-f006:**
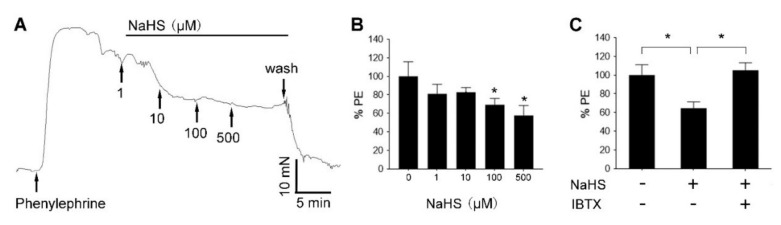
BK_Ca_ in H_2_S-induced relaxation of human uterine artery in vitro. (**A**) Freshly prepared human main uterine artery (UA) rings were preconstricted with phenylephrine (PE, 10 μM) in organ bath to achieve steady contraction for at least 5 min. Increasing concentrations (1, 10, 100, and 500 μM) of NaHS was then applied sequentially to relax the preconstricted UA ring. A representative dose–response curve of H_2_S-induced UA relaxation was shown to represent similar results of three UA ring preparations from three patients. (**B**) Bar graph summarizing the effects of NaHS on human UA (hUA) relaxation. NaHS at 100 and 500 μM decreased the artery tension to 69.3 ± 6.6% and 57.6 ± 10.8% of the maximum contraction of PE. * *p* < 0.05 compared with NaHS at 0. (**C**) NaHS (100 μM) decreased artery tension to 64.6 ± 6.7% of the maxi contraction induced by PE, and the NaHS-induced UA relaxation was reversed by co-incubation with the BK_Ca_ channel blocker iberiotoxin (IBTX, 100 nM). * *p* < 0.05.

**Figure 7 antioxidants-09-01127-f007:**
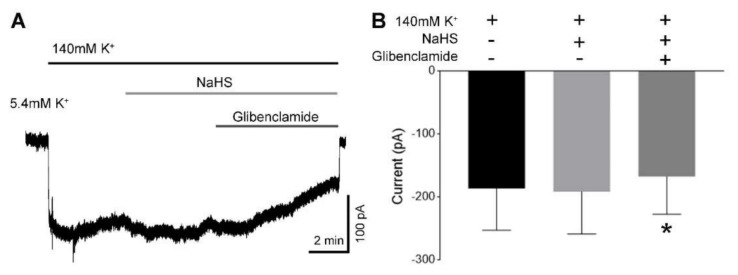
H_2_S on K_ATP_ channel activity in hUASMC. (**A**) K_ATP_ channel currents were recorded with symmetrical 140 mM K^+^ with 0.3 mM ATP in the pipette solution and the membrane potential was held at −60 mV. NaHS (300 μM) did not affect the inward K^+^ currents. (**B**) Co-incubation with the K_ATP_ channel blocker glibenclamide (10 μM) inhibited the inward currents significantly, as shown in (**B**). * *p* < 0.05 vs. baseline current induced by 140 mM K^+^ without NaHS and glibenclamide. *n* = 3/group.

**Table 1 antioxidants-09-01127-t001:** Primers used for detecting human large conductance Ca^2+^-activated voltage-dependent potassium (BK_Ca_) channel subunits by RT-PCR.

Subunits	Forward (5′-3′)	Reverse (5′-3′)	Amplicon (bp)
α	CTTCGTGGGTCTGTCCTTCC	TCTCTCGGTTGGCAGACTTG	98
β1	AAGTGCCACCTGATTGAGACC	CACAGGCATGGGTACTGGG	80
β2	GCACCGGATCGCTGTCATTA	TGGCAAAAAGACCTCCGGTA	76
β3	GAGAGGACCGAGCCGTGATG	CACCACCTAGCAGAGTCAGTGAAG	513
β4	GCGTTCTCATTGTGGTCC	TTCCAGTTGTGCCTGTTTC	243
γ1	CGCGTCAGAGGCCGAG	TGGCTAAAGGCGGCGTC	90
γ2	TCCTGGACTTCGCCATCTTC	TCAGCTCTGTGGGCTCCAC	81
γ3	TTGGGGCTCAACCCTAACAC	GAATTCCAGGGCCCCACTAC	98
γ4	TGGATCCAGGAGAACGCATC	TATCCTCCTGCTCTCCATGGG	87

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
