# Peer review of "Hydrogen Sulfide Relaxes Human Uterine Artery via Activating Smooth Muscle BKCa Channels"

_antioxidants, 2020, doi:10.3390/antiox9111127_

Round 1
Reviewer 1 Report
In this manuscript by Li et al. entitled “Hydrogen Sulfide Relaxes Human Uterine Artery via Activating Smooth Muscle BKCa Channels” the authors studied the contribution of BKCa channels in the H2S-dependent vasodilation of pregnant human uterine arteries (UA). The present work shows that BKCa channels and their auxiliary subunits (beta and gamma) are expressed in isolated human UA and primary cultured uterine artery smooth muscle cells (hUASMC) and localized mostly in the smooth muscle of the UA. The authors used electrophysiological approaches to determine the functional activation of BKCa channels in hUASMC in the absence or presence of H2S, they observed that H2S evoked an increase in current density and open probability of the channels, an effect that was partially reduced by a BKCa blocker (TEA). The H2S-dependent increase of BKCa activation was independent of extracellular Ca2+ or L-type Ca2+ channel-dependent Ca2+ entry and incubation with a reducing agent, DTT, reduced the H2S-dependent increase in BKCa currents, suggesting a role of redox signaling in this effect. Finally, using UA rings ex vivo, the authors showed that H2S-elicited vasodilation is reduced by blocking BKCa channels (IBTX). This work highlights an important mechanism for the regulation of UA vasodilation during pregnancy using human vessels and primary cultures, increasing our understanding of the role of H2S in this process. The work is well presented, and the results mostly support the conclusions. However, some key concerns may weaken the manuscript’s impact.
Major Comment:
- One conclusion of this paper is that the H2S effect on BKCa currents is redox-sensitive, however, there is a lack of proper controls in these experiments. For instance, the effect of DTT alone was not presented, this is a major concern because if DTT reduces BKCa currents in the absence of H2S, the conclusion that the H2S effect is dependent on redox state would not be supported by the results. The authors should include this control.
Minor Comments:
- It is not clear what the authors mean by ‘rings responding well to phenylephrine’ (line 205). Please clarify what measurable criteria were used to determine ‘good-responding’ rings.
- The rationale for presenting Western blots for gamma-1 subunit in Figure 1B and immunohistochemical measures of gamma-3 subunit in Figure 1C is not clear. Authors should include results for both gamma subunits using both approaches or clearly describe the rationale for not including one or the other.
- Not clear whether passages (p1, p2, p3) in Figure 1B correspond to cells from the same woman and why they were used, please clarify in the text.
- The scale bar in Figure 1C is surprisingly small (0.2 μm), this makes the UA diameter ~1 μm which seems extremely small for pregnant human UA, particularly as compared to other publications from the same research group (ref. 12).
- For consistency, make all Popen with the ‘P’ but not the ‘open’ italicized, multiple formats can be found throughout the text (i.e., lines 283 and 290).
- Lines 282-284: The statement is not clear, first it says that Popen is increased by H2S (without referring to Figure 3E), then on the same sentence it says that outwards currents were sensitive to TEA (from Figure 3A-C), please rearrange or separate these seemly unconnected statements.
- There is no reference to Figure 3D in the text. I addition, the legend of Figure 3 says ‘D&E: Representative outside-out single channel currents of BKCa control, NaHS, and IBTX groups…”, which is true for D but not for E. A more accurate recordings should be included showing baseline, NaHS, and wash. Single-channel recordings with IBTX are irrelevant in this figure if presented alone (instead of NaHS+IBTX) and without corresponding quantification.
- The justification for the use of a voltage ramp and cumulative Popen in figure 4A, and its significance, need to be addressed.
- In Figure 4D, it is hard to believe that Popen at +10 mV to +80 mV are significantly different between Control+EGTA and NaHS+EGTA, please check your statistical analysis and re-describe your conclusions if needed.
- Lines 316 and 319: Authors referred to an increase in Popen (line 316) or no effect in Popen (line 319) from results showed in Figures 4D and E, respectively. However, the graphs show current density, no Popen.
- The statement 'DTT transiently potentiated the NaHS-induced BKCa activation' (line 339) is not supported by the observations presented (Figure 5C), nor its significance discussed.
- Please expand the description of NaHS+IBTX co-incubation of the vessels. For example, why were vessels co-incubated instead of pre-incubated with IBTX and then adding NaHS? A justification for this experimental design should be included.
- In the Discussion (line 415) the authors indicate that there are 4 alpha subunits for the BKCa channel, this is incorrect, there is only one alpha subunit encoded by the gene KCNMA1.
- Line 472, the authors mentioned that they show pressurized human UA, but their results are in UA rings mounted onto a tension transducer, please correct.
- Both Y-axes in Figure 1C have misspelled the word ‘Protein’.
- In some places ‘open probability’ is written as ‘open possibility’, please correct (lines 246, 262, 321, 345).
- Line 246: replace ‘BK’ for ‘BKCa’.
- Line 283: ‘induced’ is misspelled.
- Lines 313-315: include the holding potential at which these increasing Ca2+ experiments were made.
- Line 315: Replace ‘Fig. 5C’ by ‘Fig. 4C’.
- Line 327: Replace ‘control’ by ‘baseline’.
- Line 328: Replace ‘Popen’ by ‘current density’.
Author Response
Thank you very much for your detailed and constructive comments. We revised the manuscript accordingly as below:
Major Comment:
- One conclusion of this paper is that the H2S effect on BKCa currents is redox-sensitive, however, there is a lack of proper controls in these experiments. For instance, the effect of DTT alone was not presented, this is a major concern because if DTT reduces BKCa currents in the absence of H2S, the conclusion that the H2S effect is dependent on redox state would not be supported by the results. The authors should include this control.
A: We did DTT alone group for controls, which is now added in Figure 5. DTT had no effect on BKCa single channel activity by itself, supporting that H2S augmentation of BKCa is redox-sensitive. The additions and changes in descriptions can be found in line 350-351, the figure legend of Figure 5 in Line 360.
Minor Comments:
- It is not clear what the authors mean by ‘rings responding well to phenylephrine’ (line 205).
Please clarify what measurable criteria were used to determine ‘good-responding’ rings.
A: This has been clarified in line 203 to clarify.
- The rationale for presenting Western blots for gamma-1 subunit in Figure 1B and immunohistochemical measures of gamma-3 subunit in Figure 1C is not clear. Authors should include results for both gamma subunits using both approaches or clearly describe the rationale for not including one or the other.
A: In this paper, we intended to show which subunit(s) are expressed in the uterine artery. The γ subunits are relative novel in uterine artery and we tested two commercial antibodies from Invitrogen (PA5-38058) targeting γ1 subunit and Abcam (ab121412) targeting γ3 subunit. We found that the γ1 is only detectable by Western Blot, but not by immunohistochemistry or immunofluorescence microscopy, whereas γ3 was not detectable by Western Blot. mRNA for γ2 was also detected by RT-PCR, but γ2 antibody is unavailable commercially at the moment. Thus, we showed the data with what we got with different methods. Nonetheless, this is an ongoing project that needs a lot more experiments to get the many questions addressed as we discussed, such as the role of each subunit. We continue to work on this when better antibodies becoming available.
- Not clear whether passages (p1, p2, p3) in Figure 1B correspond to cells from the same woman and why they were used, please clarify in the text.
A: The samples were from SMCs from subjects with different passages to determine if BKCa channel protein levels will change during cell passage. We add a description in line 235-236.
- The scale bar in Figure 1C is surprisingly small (0.2 μm), this makes the UA diameter ~1 μm which seems extremely small for pregnant human UA, particularly as compared to other publications from the same research group (ref. 12).
A: Sorry for this mistake. The scale bar is 0.2 mm. We have corrected in Figure 1.
- For consistency, make all Popen with the ‘P’ but not the ‘open’ italicized, multiple formats can be found throughout the text (i.e., lines 283 and 290).
A: We checked through the manuscript and reformed all the Popen to Popen with only ‘P’ italicized, which is most commonly used.
- Lines 282-284: The statement is not clear, first it says that Popen is increased by H2S (without referring to Figure 3E), then on the same sentence it says that outwards currents were sensitive to TEA (from Figure 3A-C), please rearrange or separate these seemly unconnected statements.
A: We rearranged the sentences in line 288-291 to clarify this.
- There is no reference to Figure 3D in the text. I addition, the legend of Figure 3 says ‘D&E: Representative outside-out single channel currents of BKCa control, NaHS, and IBTX groups…”, which is true for D but not for E. A more accurate recordings should be included showing baseline, NaHS, and wash. Single-channel recordings with IBTX are irrelevant in this figure if presented alone (instead of NaHS+IBTX) and without corresponding quantification.
A: Yes, you are correct. We added the text for Figure 3 D and E in line 293. We used the wrong figure for D, it should be the representative traces for E and it was corrected in the new Figure 3. The recordings are baseline, NaHS, and followed by washout as interpreted in figure E. These have been corrected in Figure 3 on page 10, and the figure legend in line 309-310 in the current version.
- The justification for the use of a voltage ramp and cumulative Popen in figure 4A, and its significance, need to be addressed.
A: Voltage ramp for single channel recording has been used in several previous studies to represent the channel properties in response to stimuli. Here we used the voltage ramp to show single channel Popen in response to different voltages in control and NaHS groups. The Popen at each membrane potential are shown on Figure 4B. The cumulative density function of opening event (P(t)) instead of cumulative Popen, would be better to describe single channel events, which is not suitable for this case since it is impossible to plot P(t) to voltage ramp before. We used Popen at each membrane potential for statistics in this study. Please see line 319-320 and 332-333.
- In Figure 4D, it is hard to believe that Popen at +10 mV to +80 mV are significantly different between Control+EGTA and NaHS+EGTA, please check your statistical analysis and re-describe your conclusions if needed.
A: In Figure 4D, paired t-test was used to compare control and NaHS groups. The big error bars may be due to cell size variations. We reanalyzed the data in which cell capacitance less than 100 pF were selected for analyzing. Please check the new uploaded figure 4D on page 16.
- Lines 316 and 319: Authors referred to an increase in Popen (line 316) or no effect in Popen (line 319) from results showed in Figures 4D and E, respectively. However, the graphs show current density, no Popen.
A: Yes, you are correct and thank you for these very critical comments. It is current density instead of Popen. We corrected them into current density (see line 338).
- The statement 'DTT transiently potentiated the NaHS-induced BKCa activation' (line 339) is not supported by the observations presented (Figure 5C), nor its significance discussed.
A: We rewrote the sentence to ‘Co-incubation with DDT blocked’ in line 350 to clarify.
- Please expand the description of NaHS+IBTX co-incubation of the vessels. For example, why were vessels co-incubated instead of pre-incubated with IBTX and then adding NaHS? A justification for this experimental design should be included.
A: Yes, you are right and that was what we did but the description was wrong. The IBTX was pre-incubated for at least 5 minutes before adding NaHS. This sentence has been rewritten to correct this (see line 367).
- In the Discussion (line 415) the authors indicate that there are 4 alpha subunits for the BKCa channel, this is incorrect, there is only one alpha subunit encoded by the gene KCNMA1.
A: Thank you for the correction. We corrected this (see line 433).
- Line 472, the authors mentioned that they show pressurized human UA, but their results are in UA rings mounted onto a tension transducer, please correct.
A: Thank you again. This has been rewritten (see line 490).
- Both Y-axes in Figure 1C have misspelled the word ‘Protein’.
A: Corrected as needed in Figure 1.
- In some places ‘open probability’ is written as ‘open possibility’, please correct (lines 246, 262, 321, 345).
A: We corrected all to ‘open probability’. Please check line 250, 268, 332 and 356.
- Line 246: replace ‘BK’ for ‘BKCa’.
A: Corrected as needed (see line 250).
- Line 283: ‘induced’ is misspelled.
A: Corrected as needed (line 293).
- Lines 313-315: include the holding potential at which these increasing Ca2+ experiments were made.
A: The cell membrane potential was held at -60 mv, this is added in line 337 and 338.
- Line 315: Replace ‘Fig. 5C’ by ‘Fig. 4C’.
A: Corrected as needed in line 326.
- Line 327: Replace ‘control’ by ‘baseline’.
A: Changed as recommended in line 337.
- Line 328: Replace ‘Popen’ by ‘current density’.
A: Replaced as needed in line 338.
Reviewer 2 Report
Comments to the authors
The study titled “Hydrogen Sulfide Relaxes Human Uterine Artery via Activating Smooth Muscle BKCa Channels” has the aim to investigate the role of BKCa channels in hydrogen sulfide relaxation in human UA smooth muscle cells; the authors conclude that BKCa channels are involved and this effect seems to be independent from extracellular Ca2+.
The study is well designed and the authors used different experimental methods such as functional, biochemical and electrophysiological methods. More relevant is that the authors used not only primary cell but also human uterus artery for functional a study to support and confirm results obtained by cell culture experiments.
There are only minor concerns that needs to be addressed.
Absract
It is not clear the term “in vivo” used in the phrase (line 80): “The purposes of this study was therefore to determine which BKCa channel subunits are expressed in the UA in vivo and cultured primary human UA SM cells. It should be better “in human UA” instead of in vivo.
The authors used in some experiments NaHS at 100 μM in other 300µM, which is the rationale? In arterial rings, the author used 1, 10, 100 and 500µM; please justify the different concentration used in different experimental methods.
Figures
In general, the quality of the figures is very poor, please improve them.
Figure 6 title “BKCa in H2S-induced relaxation of human uterine artery ex vivo”, the term “ex vivo” usually is used when there is a treatment in vivo after that the evaluation is on tissues in vitro, thus after treatment in vivo and it does not the case. All the experiment performed by rings from human artery are in vitro and not in vivo, pleas correct it.
Figure 6 The authors reported a representative trace on a dose-response curve of H2S-induced UA relaxation that is fine. In addition, the authors stated that this trace is similar to 3 UA ring preparations from 3 different patients. Thus, should be more correct to add another panel in the same figure with the results, expressed as mean+/- se, of the concentration response curve obtained with NaHS in these 3 different experiments.
Table 1
Please, change “B4” with β4
Line 366 “Treatment with H2S (300 μM)” the author used H2S or NaHS please correct through the manuscript. The effect is obviously related to H2S released but to describe the treatment conditions the authors should report the real molecule used.
Line 358 “Increasing concentrations (0, 10, 100, and 500 μM)” “0” should be “1”
There are some typing errors such as: line 412 “ndcued” or grammatical error such as “The purposes of this study was (were), please provide a spell check of the manuscript.
Author Response
Abstract: It is not clear the term “in vivo” used in the phrase (line 80): “The purposes of this study was therefore to determine which BKCa channel subunits are expressed in the UA in vivo and cultured primary human UA SM cells. It should be better “in human UA” instead of in vivo.
A: Thank you this suggestion. This has been rewritten in line 78-80.
The authors used in some experiments NaHS at 100 μM in other 300µM, which is the rationale? In arterial rings, the author used 1, 10, 100 and 500µM; please justify the different concentration used in different experimental methods.
A: NaHS at 300 μM was used when testing whether NaHS augments KATP channel as previously reported by others. We have also tested NaHS at 100 μM on KATP activity, which showed similar results as 300 uM. We added the reference in line 384.
In general, the quality of the figures is very poor, please improve them.
A: All figures are replaced with high resolution ones.
Figure 6 title “BKCa in H2S-induced relaxation of human uterine artery ex vivo”, the term “ex vivo” usually is used when there is a treatment in vivo after that the evaluation is on tissues in vitro, thus after treatment in vivo and it does not the case. All the experiment performed by rings from human artery are in vitro and not in vivo, pleas correct it.
A: As needed, ex vivo is changed as in vitro throughout the manuscript.
Figure 6 The authors reported a representative trace on a dose-response curve of H2S-induced UA relaxation that is fine. In addition, the authors stated that this trace is similar to 3 UA ring preparations from 3 different patients. Thus, should be more correct to add another panel in the same figure with the results, expressed as mean+/- se, of the concentration response curve obtained with NaHS in these 3 different experiments.
A: A bar graph for Figure 6 A is added.
Table 1, Please, change “B4” with β4
A: Corrected.
Line 366 “Treatment with H2S (300 μM)” the author used H2S or NaHS please correct through the manuscript. The effect is obviously related to H2S released but to describe the treatment conditions the authors should report the real molecule used.
A: NaHS was used throughout the study. We corrected it to NaHS (300 μM) in line 384.
Line 358 “Increasing concentrations (0, 10, 100, and 500 μM)” “0” should be “1”
A: Corrected in line 373.
There are some typing errors such as: line 412 “ndcued” or grammatical error such as “The purposes of this study was (were), please provide a spell check of the manuscript.
A: We correct the errors like throughout the manuscript and corrected them line 78, 293 and 431.
Reviewer 3 Report
The study is well designed and elegantly described. The method and pharmacological approaches were appropriate and coherent. The statistical analysis was correct. It can add new information in this field.
Minor comments
The expression of the enzymes deputed to H2S biosynthesis should be considered in human uterine artery.
Was the endothelial integrity assessed by using a concentration-response curve of Ach in human uterine artery?
Author Response
The study is well designed and elegantly described. The method and pharmacological approaches were appropriate and coherent. The statistical analysis was correct. It can add new information in this field.
A: Thank you very much for the positive comments and agree that this adds new information to the field.
Minor comments
The expression of the enzymes deputed to H2S biosynthesis should be considered in human uterine artery.
A: Important but this has been published in our paper (Ref 12).
Was the endothelial integrity assessed by using a concentration-response curve of Ach in human uterine artery?
A: Again, thank you for this important point. We added this information in the method in line 199-200.
Round 2
Reviewer 1 Report
I am pleased with the authors' responses to my comments.